# Automated Mapping of Healthcare Concepts to a Standardized Healthcare Taxonomy

Sabbir Mollah, Mohammed Rakib & Nabeel Mohammed
North South University, Dhaka, Bangladesh
{sabbir.mollah, mohammed.rakib, nabeel.mohammed}@northsouth.edu

Mashrur Wasek, AKM Shahariar Azad Rabby🆔 & Fuad Rahman
Apurba Technologies, 440 N. Wolfe Rd., Sunnyvale, CA 94085, USA
{wasekmashrur, rabby, fuad}@apurbatech.com

## Abstract

SNOMED CT presents an opportunity for numerous research prospects to learn medical terminologies and effectively assist the diagnosis process. In this work, we propose mapping the information in medical records to paths extracted from the SNOMED CT knowledge graph. To achieve this, we have leveraged language models to extract feature embeddings from EHR and compare them against the paths found in SNOMED CT. Our motivation is that this method can significantly assist the diagnosis process, ultimately leading to better healthcare outcomes. We have used several strategies to score the paths and evaluated them with experts in this field who have provided encouraging feedback on the outcomes of this approach.

## 1 Introduction

The Systematized Nomenclature of Medicine - Clinical Terms (SNOMED CT) is a standardized way of representing clinical terminologies, descriptions, synonyms, and relationships. Wang et al. (2018); Dong et al. (2022); Kraljevic et al. (2020); Searle et al. (2022); Choi et al. (2016); Agarwal et al. (2019); Beam et al. (2018); Zheng & Cui (2018) explore various deep learning and natural language processing techniques to learn embeddings from the SNOMED CT concepts. The main contribution of this paper is the novel approach to extract and rank entire paths from the SNOMED CT graph against EHRs, as opposed to previous work that would confine themselves to node-level tasks.

## 2 Methodologies

SNOMED CT can be thought as a graph $G = \langle V, E \rangle$ where V=$\{ct_1, ct_2, , ct_N\}$ is the set of all concepts, and E is the set of all relationships among the concepts. We can denote the set of all kinds of relationships as R. Finally, we can define each edge e∈E in the form e=(x,y,r) where x,y∈V and r∈R. Generating all possible simple paths from a graph is a computationally extensive problem. This is reinforced by the large number of concepts and the large number of relationships that are available in SNOMED-CT. Hence, this study only focused on generating paths for Findings concepts for hierarchical relationships to simplify the process.

We denote P as the set of all existing paths in the graph $G_{hierarchical-finding}$. So, each $p \in P$ can be written as p = $(ct_{start}, ct_i, ct_j, ct_k ct_{end})$ such that $ct_{start}$ is the top most clinical finding of $G_{hierarchical-finding}$, $ct_{end}$ is a node of $G_{hierarchical-finding}$ without any out relationships, and i,j,k∈{1,2, … N}. Our method uses a sentence encoding model M capable of taking a sentence as input and outputting a fixed-size context latent space as an embedding. For an input health record hr, an embedding h can be generated as h = M(hr).

Table 1: Most relevant paths extracted from sample medical phrases

| Input text | \|Most relevant path |
|---|---|
| Mr. Ramishen has been suffering from malaria | \|Clinical finding < General finding of observation of patient < Temperature-associated finding < Body temperature finding < Abnormal body temperature < Body temperature above reference range < Fever < Chronic fever < Intermittent fever < Malarial fever < Tertian fever < Malignant tertian fever |
| Swollen fingers were cited | \|Clinical finding < General finding of observation of patient < Swelling / lump finding < Swelling < Swelling of body structure < Swelling of body region < Swelling of limb < Swelling of upper limb < Swelling of hand < Swelling of finger < Swelling of finger joint < Finger joint - soft tissue swelling < Finger joint - synovial swelling |

Similarly, for each concept ct ∈ V, we can compute a set of embeddings C, then embedding h can be paired up with each c∈C to find a list of cosine similarity scores.

$$C = \{c_i : c_i = M(ct_i) \wedge i \in \{1, 2, ..., N\}\} \tag{1}$$

$$scores = \{cossimilarity(h, c_1), cossimilarity(h, c_2), , cossimilarity(h, c_N)\} \tag{2}$$

In our experiments, two different sentence encoding models were used to extract the embeddings of the health records and the SNOMED CT concepts, Sentence-BERT from Reimers & Gurevych (2019)'s work and all-MiniLM-L6-v2 from Wang et al. (2020)'s work. A custom dataset containing Fully Specified Names (FSNs) of concepts and their respective synonyms was created. In order to enhance the quantity and reliability of the dataset, additional paraphrased sentences were included by utilizing pre-trained paraphrasing models. As depicted in Fig. 2, when a new health record, hr, is obtained by the system, a new embedding $h$ is created.

The embedding $h$ was used to rank the path by using six different strategies. The first strategy is the aggregation of cosine similarities depicted in 2. In this strategy, if p = $(ct_m, ct_n ct_z)$ is an arbitrary path where m,n,...,z∈{1,2,3...,N}, then the score of path p will be calculated as,

$$pscore = scores[m] + scores[n] + + scores[z]. \tag{2}$$

In our observations, this strategy favors the longer sequences because of the aggregation of the scores. In another variant, the score will be divided by the number of concepts in the path, which results in shorter paths as the best matches. In our third strategy, the input embedding was contrasted against the mean of the embedding of all concepts. This approach diversified the path lengths of the best matches. Then, we have also used the Euclidean distance algorithm instead of cosine similarities in these three strategies. Finally, we have compared the strategies against our baseline TF-IDF method.

## 3   Results & Conclusion

From the results depicted in Table 1, it can be observed that the pipeline is able to identify important phrases in a sentence and match them against a relevant path. In the first example, a single mention of the term 'malaria' resulted in matching with a relevant path. A limitation that needs to be addressed in the current approach can be observed in the second example, where the model doesn't limit the "Swelling of Finger" concept and goes further in the path toward a more specific concept. To evaluate this approach, we have generated paths by using each strategies for forty different medical sentences and got them reviewed by two to experts in the field. They have qualified our method to be effective, and the claim was further fortified by making them score the paths. The results, as shown in Figure 1, indicates that our method achieves 55% satisfaction while the baseline TF-IDF approach reaches 38% satisfaction. In the future, we plan to explore faster and more efficient ways of evaluating health records and quantitatively measuring the performance of our approach.

## 4 URM Statement

The authors acknowledge that at least one key author of this work meets the URM criteria of the ICLR 2023 Tiny Papers Track.

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

## A Appendix

### A.1 Ranking algorithms

The paths for forty randomly selected medical sentences were generated using three distinct ranking methods. Each method involved calculations using both the Euclidean distance and cosine similarity algorithms. Here are the details of the methods:

1. Method 1: The similarity between the input embedding and each embedding of the concepts in a path was calculated, and the scores were aggregated.

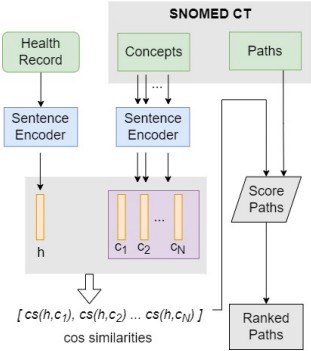

Figure 1: This figure illustrates the process of ranking the pre-extracted paths given in an EHR.

2. Method 2: The average embedding of a path was calculated, and its similarity score with the input embedding was determined.

3. Method 3: The similarity between the input embedding and each embedding of the concepts in a path was calculated, the scores were aggregated, and the aggregated score was divided by the number of concepts in the path.

In addition, the TF-IDF method was used as a baseline for extracting paths. Afterwards, paths extracted from all methods were presented to two different local domain experts. These experts voted on whether the paths were satisfactory. By combining their statements we have compiled that the cosine similarity aggregation method presented the most satisfactory results. Within the scope of our research investigation, we made an effort to reproduce the aforementioned outcomes by substituting the cosine similarity with the Euclidean distance algorithm. Regrettably, our endeavors yielded unsatisfactory results.

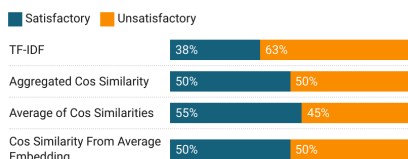

Figure 2: This figure illustrates the comparison of three scoring mechanisms using our approach against TF-IDF matching.

## A.2 Baseline

Based on our observations, while the TF-IDF approach managed to demonstrate its proficiency in simple term matching, it exhibited limitations in actually comprehending the fundamental meaning of a sentence, which is evidenced by the first example in table 2.

Table 2: Contrasting TF-IDF (Baseline) with our approach

| Input text | TF-IDF | Our approach |
|---|---|---|
| The patient's recent allergy testing confirmed an allergy to peanuts. | Clinical finding < Evaluation finding < Allergy testing - no reaction | Clinical finding < Propensity to adverse reaction < Allergy to food < Allergy to nut < Allergy to peanut |
| The patient presented with a persistent cough and shortness of breath. | Clinical finding < Respiratory finding < Respiratory function finding < Cough < Persistent cough | Clinical finding < Respiratory finding < Respiratory function finding < Cough < Dry cough < Hacking cough |

