# OpenReview forum: "Automated Mapping of Healthcare Concepts to a Standardized Healthcare Taxonomy"
_ICLR.cc/2023/TinyPapers — Submitted to Tiny Papers @ ICLR 2023_

### Official Review · Reviewer_QmHL · 2023-03-30

**Confidence:** 3

**Summary Of Contributions:**

The paper proposes to extract a SNOMET CT path for medical records using embedding similarity and path ranking.

**Rating:**

Great Start (GS): a submission which meets some of the reviewing criteria but has room for improvement

**Strengths And Weaknesses:**

Strengths:
1. The paper proposes to rank entire paths as opposed to finding relevant concepts from SNOMET.

Weaknesses:
1. It is suggested that the results are positively received by experts in the field but no details regarding the same are provided.
2. No baseline is compared (term-similarity) and it is unclear why adding scores to rank a path is a reasonable approach.

**Suggested Changes:**

It would have been useful to provide a visualization that scores multiple paths for an input text using the proposed approach and potentially contrast against simple term-matching baselines.

---

> ### Author Response · Authors · 2023-05-30
> **Added Details Regarding the Evaluation Method**
>
> We appreciate your time to review our paper. In response to your feedback, we have updated our paper accordingly.
>
> We have added details about our evaluation method in the Appendix section of the paper.
> We have used TF-IDF as a baseline model and updated it in the Appendix section. We have also experimented with different scoring algorithms other than the aggregation method.

---

### Official Review · Reviewer_jUtu · 2023-04-01

**Confidence:** 4

**Summary Of Contributions:**

The paper studies an important area of healthcare delivery by automating the mapping process of concepts in healthcare records to a standardized healthcare taxonomy.

**Rating:**

Clear, Correct, and Reproducible (CCR): a submission which meets the reviewing criteria

**Strengths And Weaknesses:**

STRENGTHS
1. The paper provides a unique method for extracting and ranking complete pathways from the SNOMED CT graph against EHRs which is a significant contribution to the healthcare field.
2. The author details the limitation of the paper and proposes grounds for improvement
3. The methodology of using sentence encoding models to generate embeddings and compute cosine similarity scores is a common practice in NLP, which adds to the reliability and reproducibility of the results.
4. The paper also provides a custom dataset of FSNs and synonyms, along with paraphrased sentences, which enhances the quantity and reliability of the dataset.

WEAKNESSES
1. The author only focus on generating paths for "Findings" concepts for hierarchical relationships, which limits the applicability of the proposed approach. Within the healthcare space, much more will be needed to provide robust and accurate matching result

**Suggested Changes:**

1. The author should consider expanding their methodology to other types of concepts and relationships in SNOMED CT, to improve the applicability of the proposed approach. SNOMED CT is robust, and even we the author do not explore all, certainly, just one really limits the usage of the approach, considering the medical sector.

---

> ### Author Response · Authors · 2023-05-30
> **More Concept and Relationship Types Not Addressed**
>
> Thank you very much for your valuable insights. We apologize for not being able to incorporate your suggestions on trying different concepts and relationships, however we have run more rigorous experiments on more scoring algorithms and included the results in the paper.

---

### Official Review · Reviewer_KgtN · 2023-04-01

**Confidence:** 4

**Summary Of Contributions:**

This paper talks about extracting paths from SNOMED CT and ranking them by scoring against electronic medical health records. The authors include a couple of examples where they generate an embedding h of the input text and rank the paths from SNOMED CT based on the aggregated cosine similarity between the path and the embedding h.

**Rating:**

Clear, Correct, and Reproducible (CCR): a submission which meets the reviewing criteria

**Strengths And Weaknesses:**

Strengths:
1. The paper is well-written, and the setup is clearly explained.
2. The premise of the paper is interesting, and it has the potential to make a significant impact with a defined problem statement.

Weaknesses:
1. More rigorous and systematic experimentation is needed in order to conclude whether scoring the paths based on the aggregated cosine similarity works efficiently.
2. Only one type of concept (Findings) and relationship (hierarchial) is considered. It would be interesting to see the analysis of multiple concepts and relationships.

**Suggested Changes:**

1. A stopping criteria can be explored in order to avoid unwanted further nodes in the path (some sort of thresholding in the cosine similarities), such that the model avoids going to the leaf nodes, that is, towards a more specific concept than required from the input text, as discussed in the second "Swollen Fingers" example.
2. More ranking algorithms, other than aggregating cosine similarity, can be explored and compared with some baselines.
3. and 4. Same as weaknesses

Minor editing: In equation (2), in c_1, 1 should be subscript

---

> ### Author Response · Authors · 2023-05-30
> **Addressed an Weakness**
>
> We deeply appreciate the time you have invested in reviewing our research paper. In our best effort we have incorporated some necessary changes according to your insightful comments and had to leave some as future work.
>
> 1. We have run more experiments trying out different variants of scoring algorithms and evaluated against a baseline TF-IDF method.
>
> 2. We couldn’t incorporate the changes on trying different types of concepts and relationships and had to leave it for future work.

---

### Meta-Review · Area_Chair_M3EQ · 2023-04-07

**Recommendation:** Invite to archive
**Confidence:** 3

**Metareview:**

The paper proposes a method to extract paths from SNOMED CT and rank them based on cosine similarity with embeddings generated from electronic medical health records. The paper is well-written and addresses an important problem in healthcare. The paper uses familiar concepts and conjoins them into a new application for medical diagnosis. The paper's custom dataset of FSNs and synonyms, along with paraphrased sentences, also enhances the quantity and reliability of the dataset.

Apart from the strengths, there are some weaknesses that should be addressed for the paper to be in an archivable state: There is a need for some additional experimentation with methodology to make a case for some choices made, and comparison against relevant baselines.
An exemplar query with the complete flow of the technique would have been very beneficial to the reader's understanding. Moreover, the claims about the impression from medical experts could be more concrete.

Overall, we recommend an invitation to archive after some revisions.

**Summary:**

The paper proposes a method to extract and rank entire paths from SNOMED CT against electronic health records using embedding similarity and path ranking. The paper is well-written and pertains to an important topic in healthcare; however, there are some additions/revisions needed to the Experiments/Results section to make the observations more solid.

**Comments And Feedback To The Authors:**

The paper presents an idea which pertains to a very important problem of reliable and efficient patient diagnosis. The core technique is a unique and novel take on how the problem has been approached as of now.

Here are some suggestions for improvement in terms of paper's robustness and clarity:
1. The authors can explore adding more ranking algorithms, other than aggregating cosine similarity alongside comparisons with some baselines to draw a clear comparison on any improvement/edge.
2. The appendix could throw some light on how the experts perceived the research tool to substantiate the positive claim.
3. It would be useful to provide a visualization that scores multiple paths for an input query using the proposed approach and potentially contrast against simple term-matching baselines.



**Reason For Not Giving A Higher Recommendation:**

As mentioned across reviews, there are some additions/revisions needed to bring the submission in an archivable state.

**Reason For Not Giving A Lower Recommendation:**

N/A

---

> ### Author Response · Authors · 2023-05-30
> **Addressed Some Weaknesses**
>
> Thank you for dedicating your valuable time to reviewing our research paper. We sincerely appreciate your feedback, and as a response, we have made the necessary revisions to address some of your suggestions.
>
> 1. We have used 7 different scoring algorithms.
>       a) Aggregated cosine similarity scores
>       b) Cosine similarity score from the average of the embedding
>       c) Averaging the cosine similarity scores
>       d) Aggregated euclidean distances
>       e) Euclidean distances from the average of the embedding
>       7) Averaging the euclidean distances
>       8) TF-IDF
>
> 2. After generating paths for 40 arbitrary medical sentences, we have consulted with two local domain experts who have carefully reviewed and scored the paths. We have updated the results in the Appendix of the paper.
>
> 3. We have set the TF-IDF method as the base line, and provided contrastive examples in the Appendix.

---

### Decision · Program_Chairs · 2023-04-10

Invite to archive

---

> ### Author Response · Authors · 2023-05-30
> **Opt-In for archival**
>
> We wish to be opted in for archival. Thank you!